# A Case of Chromosome 17q12 Deletion Syndrome with Type 2 Mayer–Rokitansky–Küster–Hauser Syndrome and Maturity-Onset Diabetes of the Young Type 5

**DOI:** 10.3390/children11040404

**Published:** 2024-03-28

**Authors:** Rosie Lee, Jung Eun Choi, Eunji Mun, Kyung hee Kim, Sun Ah Choi, Hae Soon Kim

**Affiliations:** 1Department of Pediatrics, Keimyung University Dongsan Hospital, Daegu 42601, Republic of Korea; dlfhwl@dsmc.or.kr; 2Department of Pediatrics, Ewha Womans University College of Medicine, Seoul 07804, Republic of Korea; 01247s@eumc.ac.kr (J.E.C.); 01389s@eumc.ac.kr (E.M.); 01390s@eumc.ac.kr (K.h.K.); 01175@eumc.ac.kr (S.A.C.)

**Keywords:** mullerian aplasia, maturity-onset diabetes of the young, chromosome 17q12 deletion syndrome

## Abstract

Chromosome 17q12 deletion syndrome (OMIM #614527) is a rare genetic disorder associated with a heterozygous 1.4–1.5 Mb deletion at chromosome 17q12, leading to a spectrum of clinical manifestations, including kidney abnormalities, neurodevelopmental delay, maturity-onset diabetes of the young type 5 (MODY5), and Mayer–Rokitansky–Küster–Hauser (MRKH) syndrome. We present the case of a 14-year-old Korean female diagnosed with chromosome 17q12 deletion syndrome, confirmed by chromosomal microarray analysis. The patient exhibited MODY5 with pancreatic agenesis, MRKH syndrome, dysmorphic facial features, developmental delay, kidney rotation anomaly, portal vein thrombosis with liver hypoplasia, short stature, and scoliosis. Management involved the initiation of multiple daily insulin injections for diabetes control, gynecological evaluation for MRKH syndrome, and multidisciplinary care for associated complications. This case highlights the complexity and varied organ involvement in chromosome 17q12 deletion syndrome. A comprehensive and multidisciplinary approach is crucial for the management of affected individuals, including regular monitoring, tailored interventions across various medical specialties, and providing psychosocial support.

## 1. Introduction

Chromosome 17q12 deletion syndrome (OMIM #614527) is characterized by structural or functional kidney abnormalities, maturity-onset diabetes of the young type 5 (MODY5), and neurodevelopmental disorders. The most prevalent associated features include kidney anomalies, observed in 80–90% of affected individuals. These include cystic dysplasia, hydronephrosis, vesicoureteral reflux, a single kidney, horseshoe kidney, and tubulointerstitial disease [1,2]. Developmental delay affects 50% of those affected [3]. There is increased risk of neuropsychiatric disorders, such as autism spectrum disorder and schizophrenia [4]. Various dysmorphic facial features have been noted, such as a high forehead, frontal bossing, and a depressed nasal bridge [3]. Hyperparathyroidism has been reported in up to 80% of patients with HNF1b mutations or deletions [5]. Additionally, MODY5 is present in approximately 40–45% of affected individuals [3,6]. Genital abnormalities, such as Müllerian aplasia, bicornuate uterus, uterus didelphys, are present in 30% of the affected females [3,7], while males may exhibit cryptorchidism, urethral stenosis, hypospadias, and epididymal cysts [8]. Structural and functional abnormalities of the liver [9,10], hypermetropia [4], strabiZsmus, as well as structural and exocrine abnormalities of the pancreas [11,12], are frequently observed. Less common features include congenital cardiac anomalies, musculoskeletal problems, gastroesophageal reflux disease, and seizures [3]. The diagnosis is made in a proband with a 1.4-Mb heterozygous deletion at chromosome 17q12. The estimated prevalence ranges from 1:50,000 to 1:14,000. In the European population, the estimated prevalence is 1:50,000, while in the US, it is 1:25,000 [13]. A higher estimated prevalence of 1:4000 was reported in a population-based cohort study involving 12,252 mother–father–newborn trios from Norwegian families [14]. There are currently no reports on the estimated prevalence in Korean or Asian countries.

We report a Korean case of chromosome 17q12 deletion syndrome confirmed by a 1.5-Mb deletion at chromosome 17q12. The patient displayed a phenotype characterized by MODY5, MRKH syndrome, kidney rotation anomaly, developmental delay, dysmorphic facial features, portal vein thrombosis with liver hypoplasia, short stature, and scoliosis. The systematic investigations and treatment provided to the patient are described.

## 2. Case Report

A 14-year-old female was transferred to the Department of Endocrinology for the evaluation of diabetes mellitus. The patient experienced polydipsia and polyuria for approximately 1 year without losing weight. Her height was 141.6 cm (z-score: −3.3), body weight was 39.1 kg (z-score: −1.45), and body mass index (BMI) was 19.5 kg/m^2^ (z-score: −0.35). Physical examination revealed dysmorphic facial features, including low-set ears, hypertelorism, and a prominent forehead. The Tanner stage was B4 PH5 A5 without menarche. The patient was born at 30 weeks of gestational age with a birth weight of 1.5kg because of a premature rupture of the membrane. The patient had a history of global developmental delays. Her mother and maternal grandfather had a history of diabetes mellitus and were taking oral hypoglycemic agents.

On admission, the random blood glucose level was 464 mg/dL, and HbA1c was 19.9%, without ketoacidosis. Fasting serum insulin was 3.2 uIU/mL (normal reference: 2.5–25 uIU/mL) and c-peptide was 1.4 ng/mL (reference: 1.1–4.4 ng/mL). Pancreatic autoantibodies (glutamic acid decarboxylase and insulin antibody) were negative. Abdominal computed tomography (CT) revealed dorsal pancreatic agenesis, uterine agenesis, rotation anomaly of the right kidney, chronic left portal vein thrombosis with left lobe hypotrophy of the liver, and scoliosis. Figure 1a shows dorsal pancreatic agenesis in the abdominal CT scan. Karyotype analysis was performed to rule out Turner syndrome, considering the patient’s short stature and primary amenorrhea. The results revealed 46,XX. Chromosomal microarray analysis was performed using the SNP array method (CytoScan Dx Assays, Genome build: hg19). A 1.5–Mb heterozygous deletion at 17q12 (arr[hg19] 17q12 (34,822,465–36,283,612)x1) was identified. This led to the diagnosis of chromosome 17q12 deletion syndrome. The main OMIM genes involved were HNF1B, LHX1, ZNHIT3, PIGW, AATF, TADA2A, DUSP14, DDX52, ACACA, SYNRG, and GGNBP2. The patient manifested diabetes mellitus, uterine agenesis, and developmental delay, consistent with the genetic diagnosis.

Multiple daily insulin (MDI) injections were started, and blood glucose was well controlled with a total daily insulin dose of 0.84 IU/kg/day. A complication assessment for diabetes mellitus was performed. Non-proliferative diabetic retinopathy was identified in the right eye. The patient required ophthalmological follow-up along with strict diabetes control. Nerve conduction velocity testing revealed peripheral polyneuropathy; however, the patient denied neurotic pain. The spot urine albumin-to-creatinine ratio was high at 142.9 mg/g (reference range: <30 mg/g) without microscopic proteinuria. Follow-up urinalysis will be required. Serum magnesium level was as low as 1.32 mg/dL (normal reference: 1.6–2.6 mg/dL), suggesting the possibility of renal wasting.

Gynecological examination was performed by a gynecologist. On physical examination, a micro-perforate hymen and underdeveloped clitoris were noted. Pelvic magnetic resonance imaging (MRI) revealed an absent uterus and upper vagina. Figure 1b shows the patient’s pelvic MRI. Bilateral uterine buds and triangular soft tissue in the pelvic cavity were noted, suggesting Müllerian remnants. The patient was diagnosed with MRKH syndrome. Both ovaries with follicles were also noted. The results of the hormone tests indicated luteinizing hormone of 3.5 mIU/mL, (reference range: 0.4–11.7 mIU/mL), follicle-stimulating hormone of 4.2 mIU/mL, (reference: 1.0–9.2 mIU/mL), and estradiol of 39 pg/mL (reference: 34–170 pg/mL). Surgery or vaginal dilatation therapy will be required when the patient reaches adulthood.

The patient’s short stature was evaluated. Her bone age was 16 years old according to the Greulich and Pyle method. IGF-1 level was 414 (−SD~mean for her age), and thyroid function tests were within normal range. On plain anterior-posterior imaging of the entire spine, scoliosis was observed at apex T12, convex to the left, with a Cobbs angle of 20.93°, and at apex L3, convex to the right, with a Cobbs angle of 21.32°. Consultation at the Rehabilitation Medicine Department was performed. The patient received Schroth exercise and lumbar stabilization exercise education. Brain MRI was performed to assess the pituitary gland and investigate global developmental delay. No specific focal lesions suggesting pituitary microadenoma were observed in the pituitary gland. However, a severe degree of diffuse brain atrophy, accompanied by enlargement of the lateral ventricle, was observed. The patient’s full-scale intelligence quotient was measured at 85, indicating intellectual disability. The patient did not exhibit any neuropsychiatric disorders. An assessment was performed for the incidentally discovered chronic portal vein thrombosis and liver hypoplasia during CT. No evidence of hematological disorders was observed. Factor V Leiden deficiency, factor V G1691A mutation, protein C or S deficiency, antiphospholipid antibody syndrome, F2 G20210A mutation, and myeloproliferative disorder were ruled out. Liver enzymes, including aspartate aminotransferase and alanine aminotransferase, were within the normal range, measuring 15 and 16 IU/L, respectively.

The most recent physical examination at the age of 14 years and 8 months revealed a height of 141.6 cm (z-score: −3.38), body weight of 47.7 kg (z-score: −0.52), and BMI of 24 kg/m^2^ (z-score: 1.21). The patient is on MDI injection and continuous glucose monitoring, and the last HbA1c level was 6.7% without frequent hypoglycemic events. The patient is receiving routine care in Pediatric Endocrinology, Rehabilitation Medicine, Neurology, Gynecology, Ophthalmology, and Hepatology Departments.

## 3. Discussion

We report a case of chromosome 17q12 deletion syndrome presenting with MRKH syndrome, MODY5 with pancreatic agenesis, kidney rotation anomaly, liver hypoplasia, developmental delay, short stature, and scoliosis.

MRKH syndrome affects approximately 1 in 5000 women. It is characterized by uterovaginal atresia in an otherwise typical 46,XX female. The degree of uterovaginal atresia varies from upper vaginal atresia to total Müllerian agenesis. Women with MRKH syndrome have normal ovarian development. Therefore, typical female secondary sex characteristics are preserved. MRKH syndrome is categorized into two types. Type I is characterized by isolated uterovaginal agenesis without any associated extra genital malformations. In contrast, type II MRKH syndrome includes cases with associated extragenital abnormalities, such as renal or skeletal problems. The genetics of MRKH syndrome remain unclear [15]. In a large cohort with Müllerian aplasia, 6% of women had 17q12 deletion [16]. To date, >50 genes have been proposed as candidates for MRKH syndrome, including *HNF1B*, *LHX1*, and *WNT4* [15].

Our patient had a 1.5-Mb heterozygous deletion at 17q12 (arr[hg19] 17q12(34,822,465–36,283,612) × 1) confirmed by chromosomal microarray. The OMIM genes involved were *ZNHIT3*, *PIGW*, *GGNBP2*, *LHX1*, *AATF*, *ACACA*, *TADA2A*, *DUSP14*, *SYNRG*, *DDX52*, and *HNF1B*. *LHX1* encodes a transcription factor necessary for the development of both the Müllerian and Wolffian ducts [17]. In mice with the loss of Lhx1 in the Müllerian duct epithelium, disruption in the elongation of the Müllerian duct and uterine hypoplasia characterized by loss of the entire endometrium were observed. Epithelial–mesenchymal interaction is crucial for uterine tissue differentiation [18]. *HNF1B* variants have also been associated with various renal and uterine abnormalities. However, their precise role in the pathogenesis of MRKH syndrome remains uncertain [19]. Thomson et al. [20] selectively ablated Hnf1b from the Müllerian duct epithelium in mice. They observed that this led to the hypoplastic development of uterine and kidney anomalies. Hnf1b-mutant mice exhibited a simple cuboidal epithelium and decreased stromal thickness, resulting in the inadequate development of endometrial glands. This phenotype is less severe than the Lhx1 conditional knockout, in which a complete absence of the entire endometrial layer is evident.

The diagnosis of MRKH syndrome may have profound psychological impact on both the patient and their family [21]. Upon receiving the diagnosis, many patients may encounter challenges related to identity, sexuality, and infertility. It is important to provide counseling and mental health support from the time of diagnosis and throughout the patient’s life. Referring patients to experts in genital malformations or psychologists should be considered. Sharing emotions and thoughts with other patients and support groups can be helpful [15]. Creating a functional neovagina and performing self-dilation using dilators is a method used to correct vaginal agenesis [22]. Since women with MRKH syndrome experience absolute uterine factor infertility, healthcare providers may offer options for motherhood. These options include legal adoptions, gestational surrogacy, and uterus transplantation [23].

A deletion of chromosome 17q12 covering 1.5 Mb, which includes *HNF1B*, leads to HNF1B-related syndrome. HNF1B-related syndrome is attributed to *HNF1B* haploinsufficiency [24]. MODY was diagnosed in 40% of a large cohort of individuals with either *HNF1B* mutations or deletions [6]. Bellanné-Chantelot et al. examined the *HNF1B* gene in 40 unrelated patients presenting with the MODY5 phenotype, and they observed a molecular alteration in HNF1B in 70% of the cases [25]. The HNF1B gene expressed in the epithelium of the pancreatic trunk plays a crucial role in the construction of multipotent pancreatic progenitors. This contributes to the development of both endocrine and exocrine functions. The lack of Hnf1b in mice leads to pancreatic agenesis [11]. Hnf1b ablated mice with pancreatic agenesis show a loss of several pancreatic genes, including Pax6 expression. This leads to dysfunction in both alpha and beta islet cells, resulting in glucose intolerance and diabetes mellitus [12,26]. Barbacci et al. conducted a functional study on diseases-causing mutations of *HNF1B*. They reported that HNF1B-related syndrome arises from either defective DNA-binding or transactivation function resulting from impaired coactivator recruitment [27]. Many individuals initially had some remaining insulin secretion at the time of diagnosis; however, 79% required insulin therapy during a 10-year follow-up period in a cohort study [6]. Receiving the diagnosis of diabetes mellitus and undergoing insulin treatment can be stressful for both children and their parents. Healthcare providers should consider children’s social adjustment and school performance. Furthermore, assessing psychological concerns and family stresses, and providing appropriate referrals to qualified mental health professionals, is important [28].

Morphological abnormalities of the kidney are present in 80–90% of patients with HNF1B-related syndrome [6]. In a cohort involving 80 children with renal cysts, hyper echogenicity, hypoplasia, or single kidneys, *HNF1B* mutation was identified in one-third of the patients [2]. Our patient also had a right kidney rotation anomaly. Strictly managing diabetes mellitus is important because many patients have underlying kidney problems, such as renal atrophy or a solitary kidney. The rate of end-stage renal disease is as high as 21% [6]. Hypomagnesemia is an additional issue that arises after a decrease in urine concentrating ability associated with tubulointerstitial disease. Hypomagnesemia could be the primary clinical manifestation of autosomal dominant tubulointerstitial kidney disease subtype HNF1B [29].

Furthermore, liver imaging and liver biopsy abnormalities are associated with HNF1B-related syndrome, at rates of 32% and 54%, respectively. Liver imaging abnormalities include biliary cysts, abnormal biliary ducts, and nonspecific abnormalities. Liver biopsy abnormalities include rarefaction of the biliary ducts, fibrosis, steatosis, and focal nodular hyperplasia [6,9]. On abdominal CT, our patient showed left portal vein thrombosis and left lobe hypoplasia of the liver. Liver enzymes, including ALT, coagulation profiles, and mutation studies for hematological disorders were normal.

Approximately half of the individuals with 17q12 deletion syndrome experience some degree of learning disability [10,30]. The risk of autism spectrum disorder, schizophrenia, and intellectual disability in patients with 17q12 deletion syndrome is higher than that in unaffected individuals [4]. Our patient exhibited intellectual disability without the presence of neuropsychiatric disorders. Children with autism spectrum disorder of intellectual disability may benefit from specialized education, and it is important for parents to be informed about the available options. These may include medical centers specializing in psychiatric disorders, autism centers, and schools tailored for learning disabilities.

Musculoskeletal disorders, such as joint laxity, pectus deformity, clinodactyly, hip dysplasia, scoliosis, and short stature have been reported in individuals with chromosome 17q12 deletion syndrome [8]. Postnatal growth reduction and short stature are known to be associated; however, the underlying etiology remains unelucidated [31]. Ai et al. reported cases of growth hormone deficiency and precocious puberty in patients with MRKH syndrome. A possible mechanism for the occurrence of precocious puberty in patients with MRKH syndrome involves a lack of communication between the ovary and uterus, leading to disruptions in the hypothalamic-gonadal feedback system. This disturbance triggers the premature secretion of gonadotropins [32]. Untreated precocious puberty can be attributed to a negative final adult height. In the case of our patient, the gonadotropin-releasing hormone stimulation test was not performed because of the patient’s late age at presentation. However, considering the advanced bone age, untreated precocious puberty cannot be ruled out. Furthermore, we did not perform a growth hormone stimulation test because the IGF-1 level was within the normal range, and there was no delay in the bone age. The chances of growth hormone deficiency were less. Skeletal abnormalities, including scoliosis, and the potential presence of untreated precocious puberty may have contributed to the short stature observed in our patient.

In conclusion, we report a case of chromosome 17q12 deletion syndrome. When a patient presents to the hospital with any of the described symptoms, chromosome 17q12 deletion syndrome should be suspected and considered. When diagnosed with this condition, a systematic workup should be performed because chromosome 17q12 deletion syndrome can affect multiple organs. For optimal patient care, a multidisciplinary team should be assembled, including endocrinologists, gynecologists, neurologists, nephrologists, hepatologists, rehabilitation medicine experts, and geneticists. Caring for patients with chromosome 17q12 deletion syndrome requires a multidisciplinary approach that addresses both physiological and psychosocial issues.

## Figures and Tables

**Figure 1 children-11-00404-f001:**
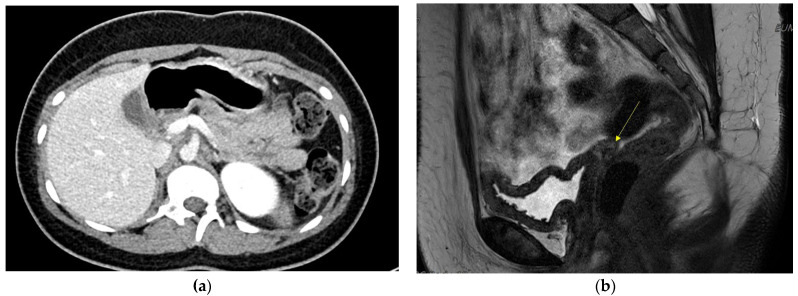
(**a**) Dorsal pancreas agenesis in Abdomen CT.; (**b**) Absent uterus and upper vagina in pelvis MRI. Yellow arrow shows absent uterus.

## Data Availability

The data is not publicly available to protect patient privacy.

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
