# Peer review of "A Case of Chromosome 17q12 Deletion Syndrome with Type 2 Mayer–Rokitansky–Küster–Hauser Syndrome and Maturity-Onset Diabetes of the Young Type 5"

_children, 2024, doi:10.3390/children11040404_

Round 1

Reviewer 1 Report

Comments and Suggestions for Authors

This study, which presents a case of Mayer–Rokitansky–Küster–Hauser syndrome, is quite remarkable. It is appreciated that this syndrome was accompanied by diabetes and that it reveals the difficulties experienced by the case. But some situations need clarification. For example, the introduction section should be strengthened by conducting a more comprehensive research on which issues these phenomena challenge children's lives the most, what disruptions occur during the developmental periods of these phenomena, and which diseases most often accompany this phenomenon.

 I also recommend that more details be discussed in the case report.

For example, in this case, diabetes was accompanied, I wonder what else?

Diseases are observed in these cases, what is their frequency?

Is there a population where it is particularly common?

Or in which countries it is seen more frequently.

These issues need to be discussed in the introduction of the article.

Also, what should be done in such cases and how does this situation affect the psychosocial and physiological life of children.

What kind of support do children need?

If the authors can add these issues to the article, the article will be much more developed.

Comments on the Quality of English Language

I particularly recommend that the paragraph in which most of the presentation is made be reconsidered. I recommend that you avoid long sentences that are difficult to understand and write your findings more concisely.

Author Response

Thank you very much for taking the time to review this manuscript. Please find the detailed responses below and the corrections highlighted in the re-submitted files.

Point-by-point response to Comments and Suggestions for Authors

Comments 1: It is appreciated that this syndrome was accompanied by diabetes and that it reveals the difficulties experienced by the case. But some situations need clarification. For example, the introduction section should be strengthened by conducting a more comprehensive research on which issues these phenomena challenge children's lives the most, what disruptions occur during the developmental periods of these phenomena, and which diseases most often accompany this phenomenon.

Response 1: Thank you for pointing this out. I agree with this comment. In the introduction, additional details regarding potential accompanying symptoms have been added into lines 33-48. 

Comments 2:  I also recommend that more details be discussed in the case report. For example, in this case, diabetes was accompanied, I wonder what else?

Response 2: In the introduction, I briefly outlined the patient’s phenotype in lines 55-60, with further details provided in the case report. (About diabetes: line 73-96, MRKH syndrome: line 97-105, short stature and scoliosis: line 106-111, developmental delay: line 112-117)

Comments 3: Diseases are observed in these cases, what is their frequency?

Response 3: I have included additional sentences in the introduction, providing detailed information about potential accompanying symptoms with the reported frequencies in lines 33-48. 

Comments 4: Is there a population where it is particularly common?
Or in which countries it is seen more frequently.

Response 4: I have included sentences in line 50-54 discussing the estimated prevalences of the condition across the world.  

Comments 5: Also, what should be done in such cases and how does this situation affect the psychosocial and physiological life of children. What kind of support do children need?

Response 5: Thank you for pointing this out. I agree with this comment. In lines 165-168, I discussed the psychosocial impact of receiving a diagnosis of MRKH syndrome, while the support required by children and later in life is described in lines 168-175.

In lines 192-196, I addressed the emotional distress associated with receiving a diagnose of diabetes and undergoing insulin treatment, emphasizing the importance of providing psychosocial support for both the children and their families. 

In lines 220-223, I addressed the necessity of specialized education support for children with developmental delay or neuropsychiatric disorders. 

As a result, I’ve included lines 248-250, emphasizing the necessity for psychosocial and physiological support for affected children. 

Response to Comments on the Quality of English Language
Point 1: I particularly recommend that the paragraph in which most of the presentation is made be reconsidered. I recommend that you avoid long sentences that are difficult to understand and write your findings more concisely.

Response 1: Thank you for pointing this out. I have gone through the manuscript from the start and revised lengthy sentences, making them more concise.

Reviewer 2 Report

Comments and Suggestions for Authors

The case study  "A case of chromosome 17q12 deletion syndrome with type 2 2 Mayer–Rokitansky–Küster–Hauser syndrome and maturity– on-3 set diabetes of the young type 5" is a well-written case study of a 14-year-old 17q12 deletion syndrome female patient with multiple abnormalities and syndromes related to the genes which are absent as a result of this chromosomal aberration. The paper looks good in its overall findings and presentation. There are, however, a few things that could be added to the paper to improve the information related to the 17q12 condition.

The only other thing I would have liked to see would be statistical data about the prevalence of this condition in Korea, PAN Asia, Europe and America to provide a glimpse of the geographical and racial distribution of this condition.

Author Response

Thank you very much for taking the time to review this manuscript. I revised my manuscript and highlighted the changed sentences. 

Point-by-point response to Comments 
Comments 1: The only other thing I would have liked to see would be statistical data about the prevalence of this condition in Korea, PAN Asia, Europe and America to provide a glimpse of the geographical and racial distribution of this condition.

Response 1: Thank you for pointing this out. I added statistical data in the introduction section (line 50-54) about the estimated prevalences in Europe and the US. However, I couldn’t find any reports about the prevalence in Asia or Korea. I think this topic warrants further investigation.